# Bevacizumab Treatment for Metastatic Colorectal Cancer in Real-World Clinical Practice

**DOI:** 10.3390/medicina59020350

**Published:** 2023-02-13

**Authors:** Ioana Mihaela Dinu, Mariana Mihăilă, Mircea Mihai Diculescu, Vlad Mihai Croitoru, Adina Turcu-Stiolica, Diana Bogdan, Monica Ionela Miron, Cristian Virgil Lungulescu, Sorin Tiberiu Alexandrescu, Traian Dumitrașcu, Florina Buică, Ioana Niculina Luca, Cristina Lungulescu, Madalina Cristina Negulescu, Iulia Magdalena Gramaticu, Irina Mihaela Cazacu, Adina Emilia Croitoru

**Affiliations:** 1Department of Oncology, Fundeni Clinical Institute, 022328 Bucharest, Romania; 2Department of Internal Medicine, Fundeni Clinical Institute, 022328 Bucharest, Romania; 3Faculty of Medicine, Carol Davila University of Medicine and Pharmacy, 050474 Bucharest, Romania; 4Department of Gastroenterology, Fundeni Clinical Institute, 022328 Bucharest, Romania; 5Department of Pharmacoeconomics, University of Medicine and Pharmacy of Craiova, 200349 Craiova, Romania; 6Department of Oncology, University of Medicine and Pharmacy Craiova, 200349 Craiova, Romania; 7Department of Oncology, County Clinical Emergency Hospital, 200642 Craiova, Romania; 8Department of Surgery, Fundeni Clinical Institute, 022328 Bucharest, Romania; 9Faculty of Medicine, Titu Maiorescu University, 031593 Bucharest, Romania

**Keywords:** colorectal cancer, metastases, bevacizumab, anti-VEGF, real-world data

## Abstract

*Background and Objectives:* Colorectal cancer (CRC) is a leading cause of cancer-related mortality and morbidity worldwide. Bevacizumab was approved for the treatment of metastatic colorectal cancer (mCRC) based on favorable benefit-risk assessments from randomized controlled trials, but evidence on its use in the real-world setting is limited. The aim of the current study is to evaluate the outcomes and safety profile of bevacizumab in mCRC in a real-world setting in Romania. *Patients and Methods:* This was an observational, retrospective, multicentric, cohort study conducted in Romania that included patients with mCRC treated with bevacizumab as part of routine clinical practice. Study endpoints were progression-free survival, overall survival, adverse events, and patterns of bevacizumab use. *Results*: A total of 554 patients were included in the study between January 2008 and December 2018. A total of 392 patients (71%) received bevacizumab in the first line and 162 patients (29%) in the second line. Bevacizumab was mostly combined with a capecitabine/oxaliplatin chemotherapy regimen (31.6%). The median PFS for patients treated with bevacizumab was 8.4 months (interquartile range [IQR], 4.7–15.1 months) in the first line and 6.6 months (IQR, 3.8–12.3 months) in the second line. The median OS was 17.7 months (IQR, 9.3–30.6 months) in the first line and 13.5 months (IQR, 6.7–25.2 months) in the second line. Primary tumor resection was associated with a longer PFS and OS. The safety profile of bevacizumab combined with chemotherapy was similar to other observational studies in mCRC. *Conclusions*: The safety profile of bevacizumab was generally as expected. Although the PFS was generally similar to that reported in other studies, the OS was shorter, probably due to the less frequent use of bevacizumab after disease progression and the baseline patient characteristics. Patients with mCRC treated with bevacizumab who underwent resection of the primary tumor had a higher OS compared to patients with an unresected primary tumor.

## 1. Introduction

Colorectal cancer (CRC) is a leading cause of cancer-related mortality and morbidity worldwide and represents a significant public health challenge in Romania [1]. According to the GLOBOCAN database, CRC is the most frequent type of cancer in Romania [2,3]. The diagnosis of CRC is often delayed due to the limited availability of screening programs and low public awareness about the importance of early detection. Despite improvements in colorectal mortality rates in most European countries over the last decade, the survival rates in Romanian patients with CRC are consistently lower [1].

Initial clinical presentation as metastatic CRC (mCRC) occurs in approximately 20% of patients. Furthermore, up to 50% of patients with localized disease will develop metastases [4]. Curative resection is only possible in a small proportion of metastatic patients with limited disease. Palliative systemic chemotherapy is the most commonly used treatment modality in order to improve overall survival (OS) while maintaining quality of life [4,5]. Various combinations of chemotherapy have been studied for the treatment of mCRC, and the addition of molecularly targeted therapies to chemotherapy as well as the sequential use of all available treatments have contributed to a gradual improvement in survival, with the median OS currently reaching 2 to 3 years [4,5,6,7,8,9,10,11]. The treatment options typically include a fluoropyrimidine-based triplet (FOLFIRINOX), doublet (FOLFOX/CAPOX or FOLFIRI/CAPIRI), or fluoropyrimidine monotherapy (5-FU/folinic acid, or capecitabine) combined with a biological agent targeting either the vascular endothelial growth factor (VEGF) in an unselected population or the epidermal growth factor receptor (EGFR) in patients with RAS wild-type tumors [4,5,12].

Bevacizumab is a recombinant humanized monoclonal antibody that was approved for patients with mCRC in 2004 in combination with fluoropyrimidine-based chemotherapy [13]. This angiogenesis inhibitor prevents vascular endothelial growth factor A from binding to its receptors, VEGFR-1 and VEGFR-2, leading to tumor growth inhibition [6,14]. In June 2006, the US Food and Drug Administration (FDA) extended bevacizumab’s approval for the second-line treatment of mCRC patients [9,15]. Moreover, continuation of VEGF inhibition with bevacizumab plus standard second-line chemotherapy beyond disease progression had clinical benefits in patients with mCRC, according to a randomized phase III clinical trial [16]. Biosimilars for bevacizumab have also been recently approved by the FDA and the European Medicines Agency (EMA).

CRC was the first malignancy for which clear evidence for the efficacy of an anti-VEGF strategy was demonstrated in randomized trials [17]. In a pivotal early trial, the addition of bevacizumab to the bolus IFL (irinotecan, 5-FU, and leucovorin) regimen significantly improved response rates, time to tumor progression, and OS [13]. Since then, the benefit of adding bevacizumab to a variety of chemotherapy regimens as first-line therapy has been confirmed, although the magnitude of both the OS and PFS benefits were relatively modest [17]. A pooled analysis of trials comparing chemotherapy with and without bevacizumab in the first-line setting showed that the addition of bevacizumab was associated with a significant reduction in the risk of death but with a modest advantage in OS and PFS of only 2 months [18]. These issues have led to a debate regarding the routine use of bevacizumab as a component of first-line therapy in patients with unresectable mCRC. There might be subgroups of patients for whom the benefit of bevacizumab does not outweigh its risks; however, this remains a controversial area. Bevacizumab is typically well tolerated, but it can be associated with adverse events such as hypertension, thromboembolic and bleeding events, wound healing complications, and gastrointestinal perforation [13,19,20,21]. Although these are potentially serious outcomes, they are not common.

Bevacizumab was approved for the treatment of metastatic colorectal cancer (mCRC) based on favorable benefit-risk assessments from randomized controlled trials [9,13,22,23], but evidence on its use in the real-world setting is limited. Most randomized clinical trials include only 10% of “real-world” patients [24]. The remaining 90%, including those with significant comorbidities, those living in remote regions, and elderly patients, are underrepresented in clinical trials [24,25]. However, there are several observational studies that provide real-world data to support the use of bevacizumab in combination with chemotherapy in patients with mCRC. These include the Avastin Registry—Investigation of Effectiveness and Safety (BRITE) study [26] and the Bevacizumab Regimens: Investigation of Treatment Effects and Safety (BRiTE) study [21], both conducted in the United States. The Bevacizumab Expanded Access Trial (BEAT) [27] was another observational, multicentric study evaluating the safety and efficacy of bevacizumab plus first-line chemotherapy in a general cohort of patients with mCRC. The Avastin ColORectal Non-interventional (ACORN) study [6] assessed the outcomes and safety of bevacizumab in a real-world setting in the United Kingdom, while the CONCERT study [15] addressed French specificities regarding the use of bevacizumab in mCRC patients. The non-interventional study KORALLE provided broad real-world evidence on the effectiveness and safety of bevacizumab in the German population [28]. However, results from these studies may not be entirely applicable to Romanian practice.

The aim of the current study is to evaluate the efficacy outcomes and safety profile of bevacizumab in mCRC in a real-world setting in Romania.

## 2. Patients and Methods

This was an observational, retrospective, multicentric cohort study conducted in Romania that included patients with mCRC who were treated with bevacizumab at the Fundeni Clinical Institute, Bucharest, and the Oncolab, Craiova, between 2008 and 2018. Access to patients’ data was approved by the local committee of the hospitals. The data was collected and stored while maintaining complete anonymity. The study was in accordance with the Declaration of Helsinki and its amendments.

Patients with mCRC who had received bevacizumab as part of their treatment and were over the age of 18 were eligible. The physician decided on the chemotherapy regimen, as well as the dose and frequency of bevacizumab administration. Bevacizumab was started at the same time as or within three months of the first-line chemotherapy regimen. Bevacizumab administration was delayed until 2016 as a result of a National House of Insurances committee approval.

The study required the collection of data that are usually assessed during the management of mCRC and were available in the medical records. Patients’ characteristics (age and gender), disease characteristics (date of diagnosis, tumor location, and stage), data on treatments received (date, dose, changes, discontinuation, reason for change, and discontinuation), progression of disease (date), and death (date and cause) were collected. Safety outcomes focused on previously described adverse events related to bevacizumab [9,13]. Data collected included bevacizumab-related adverse events (AEs) and serious adverse events (SAE). Adverse events of special interest included proteinuria, hypertension, bleeding, bowel perforation, impaired wound healing, arterial thromboembolic events, and reversible posterior multifocal leukoencephalopathy.

### 2.1. Study Endpoints

The primary endpoints of the study were progression-free survival (PFS) and overall survival (OS). The PFS time was defined as the time from the start of therapy to disease progression or death from any cause in the study. The OS time was defined as the time from the start of therapy to death from any cause.

Secondary endpoints were patient and disease characteristics; patterns of bevacizumab use (dose, duration, and combined chemotherapy); rate of response; and adverse events.

### 2.2. Statistical Analysis

Total numbers and percentages were used to represent categorical variables, and, for continuous variables, the mean (standard deviation) and median (interquartile range) were used if they did not follow a normal distribution. We compared the patients’ characteristics and the endpoints (progression-free survival, PFS, and overall survival, OS) for the two groups of patients (group I, patients with mCCR treated with bevacizumab in the first line, and group II, patients with mCCR treated with bevacizumab in the second line). A Mann–Whitney test was used for the continuous variables and the chi-square or Fisher’s exact test for the categorical variables. The log-rank test was used for PFS and OS to assess the differences between the treatment groups. Time-to-event endpoints were analyzed using the Kaplan–Meier methods. A univariate analysis was used to identify the prognostic factors for survival in the first and second lines. The factors with a *p*-value of less than 0.05 were incorporated into the subsequent multivariable analysis using a Cox proportional hazards regression model. Data were analyzed using GraphPad Prism 9.0.0 software (GraphPad Software, LLC, San Diego, CA, USA). A two-sided *p*-value smaller than 0.05 was statistically significant.

## 3. Results

A total of 554 patients were included in the study between January 2008 and December 2018. Bevacizumab was administered to 392 patients (71%) in the first line, while 162 patients (29%) received bevacizumab in the second line. Baseline patients’ characteristics are summarized in Table 1. The mean age was 61 (range: 20–84 years), with 56%, 4%, and 1% of patients aged ≥65, ≥75, and ≥80 years, respectively. Synchronous metastases were reported for 380 (68.6%) patients (Appendix A). At inclusion, 431 (77.8%) patients had liver metastases, and 154 (27.8%) had more than two metastatic sites. The primary tumor was located in the colon for 74.2% of patients and in the rectum for 25.8%. Overall, 454 (81.9%) patients underwent primary tumor resection.

### 3.1. Pattern of Bevacizumab Treatment

The median duration of bevacizumab treatment was 6.7 months. Treatment with bevacizumab was most frequently combined with capecitabine/oxaliplatin (CAPOX) (31.6%), followed by fluorouracil/folinic acid/oxaliplatin (FOLFOX) (24.2%), and fluorouracil/folinic acid/irinotecan (FOLFIRI) (20.9%) (Table 2, Appendix A). Overall, a higher percentage of patients in the oxaliplatin–bevacizumab subgroup were exposed to all three active chemotherapies (i.e., 5-FU, oxaliplatin, and irinotecan) in the metastatic setting compared to patients in the irinotecan–bevacizumab subgroup (40% vs. 21%, respectively).

The most common reasons for bevacizumab discontinuation were progressive diseases (87%). Other documented reasons included treatment toxicity or clinical deterioration. A total of 161 patients (29%) received bevacizumab after disease progression.

### 3.2. Efficacy

The median PFS for patients treated with bevacizumab in the first line was 8.4 months (interquartile range [IQR], 4.7–15.1 months) (Figure 1). For patients treated in the second line, the median PFS was 6.6 months (IQR, 3.8–12.3 months). No differences were found in the median PFS according to the chemotherapy regimen, the resection status of the primary tumor, the location of metastases, the number of metastatic sites, the type of metastases, or the RAS status. For the second-line group, patients with primary tumor resection (*p* 0.049) and RAS mutations (*p* 0.006) had statistically longer median PFS (Appendix A).

The median OS for patients treated with bevacizumab in the first line was 17.7 months (IQR, 9.3–30.6 months), while the median OS for patients treated in the second line was 13.5 months (IQR, 6.7–25.2 months) (Figure 1). For the first-line group, no significant difference in the median of OS was found according to the chemotherapy chosen, K-Ras status, or location of metastases (isolated liver metastases vs. other). However, patients with a primary tumor resection (*p* 0.005), left-sided tumors (*p* 0.003), and metachronous metastases (*p* 0.01) had a longer median OS (Appendix A).

The rate of response (complete response and partial response) for bevacizumab/oxaliplatin-based chemotherapy in the first line was 20.8% and 23% for bevacizumab/irinotecan-based chemotherapy. In the second line, the rate of response was 8.6% for bevacizumab/oxaliplatin-based chemotherapy and 12.7% for bevacizumab/irinotecan-based chemotherapy (Table 3).

A Cox hazard regression was performed to identify the factors affecting the PFS or OS. In univariate analysis, no factor was found to be a statistically significant predictor of poor outcomes for PFS. Conversely, location of the tumor (right vs. left) and resection status of the primary tumor (yes vs. no) were prognostic factors for OS with HR 1.54 (1.09–2.16), *p* 0.014, and HR 1.02 (1.003–1.03), *p* 0.016, respectively (Table 4).

### 3.3. Safety

Safety data were not available for 3% (17 patients) of the study population. A total of 295 patients (55%) experienced at least one AE related to bevacizumab. The most frequent bevacizumab-related AEs were bleeding (25.9% of patients), hypertension (16%), proteinuria (12.4%), and venous thromboembolic events (7%). Forty-eight patients (8.9%) experienced grade 3–4 bevacizumab-related AEs. The most common grade 3–4 AEs were hypertension (2%), venous thromboembolic events (2%), fistula (1.3%), gastrointestinal perforation (1%), bleeding (0.9%), and arterial thromboembolic events (0.9%) (Table 5). No unforeseen related AE was reported.

## 4. Discussion

This retrospective, observational study provided a comprehensive analysis of treatment patterns, efficacy, and safety of bevacizumab in mCRC in a real-world setting in Romania. The study assessed the clinical outcomes associated with bevacizumab in a large, relatively unselected, and general clinical practice population.

Most patients included in the study (71%) initiated bevacizumab in the first line. CAPOX was the most frequently administered first-line chemotherapy in combination with bevacizumab (31.6%), similar to the data reported by a real-world study conducted in the UK [6]. Three other observational studies reported FOLFOX as the most common chemotherapy regimen associated with bevacizumab [26,27,29]. Overall, the increased use of capecitabine did not have an impact on survival, as no difference in OS was observed between the different therapy regimens. Many oncologists choose FOLFOX in combination with bevacizumab based on the results of the US Intergroup N9741 trial, which demonstrated a significant difference in outcome favoring FOLFOX over IFL [30]. However, other studies showed only a modest benefit for adding bevacizumab to FOLFOX [9,22,31]. In the French CONCERT study [15], the combined therapy consisted mainly of FOLFIRI (68%). In our study, the rate of patients receiving FOLFIRI in the first line was much lower (24.2%).

The median PFS of 8.4 months [IQR, 4.7–15.1] reported for patients treated with bevacizumab in the first line was similar to results observed in other observational studies (8.7–10.8 months) [15,21,26,32] and in interventional clinical trials (9.3–10.6 months) [9,13,22]. The median OS of 17.7 months was consistent with the findings of the ACORN study (17.8 months) [6] but considerably shorter compared to other observational studies: BRiTE (22.9 months) [21], BEAT (22.7 months) [27], and ARIES (23.2 months) [26]. A possible explanation includes the less frequent use of bevacizumab beyond disease progression. In our study, only 29% of patients received bevacizumab after disease progression, compared to 44.4% in BRiTE [21,29] and 40.5% in ARIES [26]. However, the differences in study design and study populations may also be responsible for the differences in survival, and therefore direct median OS comparisons should be interpreted with caution.

The median PFS and OS for the second-line subgroup were consistent with the findings of other studies; the median PFS was 6.6 months, and the median OS was 13.5 months. In the interventional study by Giantonio et al. [9], the median PFS and OS were 7.3 and 12.9 months, respectively.

The median PFS and OS were consistent across the chemotherapy regimens in our study, suggesting that the effectiveness of bevacizumab is not related to the chemotherapy regimen used.

In the first line, patients with a resected primary tumor had a higher OS compared to those with an unresected primary tumor. The impact of the primary tumor resection on OS is consistent with results reported recently [15,33,34,35]. The resection of the primary tumor before systemic treatment for mCRC reduces the risk of complications requiring urgent surgery. Primary tumor resection may be beneficial as it removes the “most aggressive” source of malignant cells and proangiogenic factors [15]. A recent meta-analysis showed that mCRC patients with resected primary tumors have better survival than those without surgery of the primary tumor when treated with bevacizumab [35]. Thus, the resection status can be used as a prognostic factor to predict the beneficial effects of bevacizumab containing regimens. However, most of the available data on the impact of primary tumor resection comes from subgroup analyses or observational studies and needs to be confirmed by randomized trials [15].

The safety profile of bevacizumab was generally as expected. Grade 3–4 AE related to bevacizumab were reported for 8.9% of patients, similar to the results of other observational studies [15]. No unexpected AE were reported. However, direct comparisons of safety data with other observational studies are difficult to make considering the differences in data collection methods. For example, in some studies [6,27], all adverse events, including serious adverse events, were collected. Other studies [21,26] reported only bevacizumab-related adverse events (protocol-specified).

The limits of this large observational study are the retrospective design and the unselected and uncontrolled patient populations treated in a real-life setting. Given the retrospective design, tumor response assessment was not standardized. This may have affected the results of the PFS and the response rate, but it reflects routine clinical practice. The results should therefore be interpreted with care. It should also be noted that we collected data on bevacizumab-associated AEs, and differences in chemotherapy-related toxicities between the oxaliplatin-bevacizumab and irinotecan-bevacizumab regimens were not evaluated.

## 5. Conclusions

The purpose of the current study was to address Romanian specificities regarding the routine use of bevacizumab in mCRC patients. The safety profile of bevacizumab was generally as expected. The OS was shorter even though the PFS was generally similar to that reported in other studies. The lower-than-expected OS is probably caused by the baseline patient characteristics and the less frequent use of bevacizumab after disease progression. Moreover, there were differences in study design and study populations among studies; thus, a direct comparison of PFS and OS data should be interpreted with caution. Another important observation of the current study was that patients with mCRC treated with bevacizumab who underwent resection of the primary tumor had a longer OS compared to patients with an unresected primary tumor. However, most available data on the impact of primary tumor resection came from subgroup analyses or observational studies and therefore need to be confirmed by randomized trials.

It is important to emphasize that real-world data from studies using bevacizumab in mCRC patients can provide valuable insights into the clinical practice of oncology and help to inform treatment decisions for patients with mCRC.

## Figures and Tables

**Figure 1 medicina-59-00350-f001:**
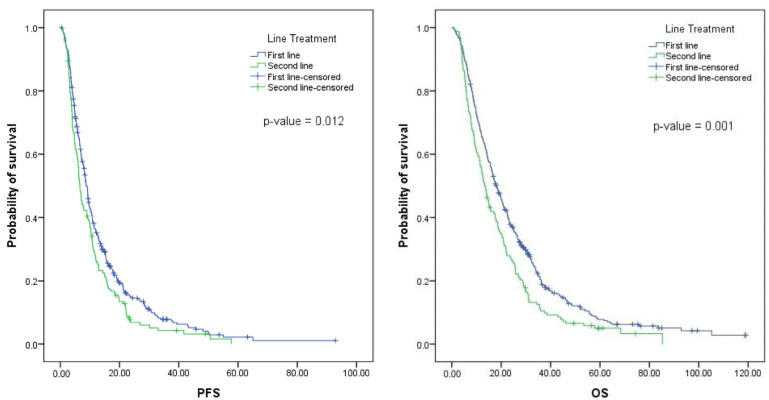
Survival curves of patients with mCCR treated with bevacizumab.

**Table 1 medicina-59-00350-t001:** Patients and disease characteristics (*N* = 554).

	Whole Group (*N* = 554)	First Line (*N* = 392)	Second Line (*N* = 162)	*p*-Value
**Age, years** mean, standard deviation median, interquartile range	59.9 (10.6) 61 (54–67)	59.7 (10.9) 61 (53–67.8)	60.5 (9.8) 61 (54–67)	0.680 ^1^
**Gender** Female Male	215 (38.8%) 339 (61.2%)	158 (40.3%) 234 (59.7%)	57 (35.2%) 105 (64.8%)	0.2920 ^2^
**Primary tumor site** Cecum	41 (7.4%)	27 (6.9%)	14 (8.6%)	
Ascending colon	43 (7.8%)	31 (7.9%)	12 (7.4%)	
Descending colon Sigmoid colon	42 (7.6%) 134 (24.2%)	30 (7.7%) 89 (22.7%)	12 (7.4%) 45 (27.8%)	
Transverse colon	26 (4.7%)	20 (5.1%)	6 (3.7%)	
Hepatic flexure Splenic flexure	30 (5.4%) 26 (4.7%)	24 (6.1%) 22 (5.6%)	6 (3.7%) 4 (2.5%)	
Recto-sigmoid junction	69 (12.5%)	48 (12.2%)	21 (13%)	
Rectum	143 (25.8%)	101 (25.8%)	42 (25.9%)	
**Primary tumor localization** Left Right	414 (74.7%) 140 (25.3%)	290 (74%) 102 (26%)	124 (76.5%) 38 (23.5%)	0.5913 ^2^
**Cancer grading** G1 G2 G3	140 (25.3% 282 (50.9%) 47 (8.5%)	88 (22.4%) 206 (52.6%) 33 (8.4%)	52 (32.1%) 76 (46.9%) 14 (8.6%)	0.0998
**Resection status of the primary tumor** Yes No	454 (81.9%) 96 (17.3%)	312 (79.6%) 77 (19.6%)	142 (87.7%) 19 (11.7%)	0.0262 ^2,^*
**Metastases resection** Yes No	131 (23.6%) 419 (75.6%)	89 (22.7%) 300 (76.5%)	42 (25.9%) 119 (73.5%)	0.4421 ^2^
**RAS** Non-mutant Mutant NA	154 (27.8%) 166 (30%) 234 (42.2%)	94 (24%) 133 (33.9%) 165 (42.1%)	60 (37%) 33 (20.4%) 69 (42.6%)	0.0009 ^3,^*

^1^ Mann–Whitney U test *p*-value; ^2^ Fisher’s exact test *p*-value; ^3^ Chi-square test *p*-value; * significant difference.

**Table 2 medicina-59-00350-t002:** Chemotherapy regimens associated with bevacizumab.

	Whole Group (*N* = 554)	First Line (*N* = 392)	Second Line (*N* = 162)	*p*-Value
Chemotherapy regimen Capecitabine MCT	16 (2.9%)	7 (1.8%)	9 (5.6%)	
Capecitabine + Irinotecan	86 (15.5%)	43 (11%)	43 (26.5%)	
Capecitabine + Oxaliplatin	175(31.6%)	158 (40.3%)	17 (10.5%)	
LV + 5-FU + Irinotecan	116 (20.9%)	57 (14.5%)	59 (36.4%)	
LV + 5-FU + Oxaliplatin	134 (24.2%)	116 (29.6%)	18 (11.1%)	
LV + 5-FU + Irinotecan + Oxaliplatin LV + 5-FU	4 (0.7%) 23 (4.2%)	4 (1%) 7 (1.8%)	0 (0%) 16 (9.9%)	
Chemotherapy regimen Irinotecan-based Oxaliplatin-based Mono fluoropyrimidine-based	202 (36.5%) 309 (55.8%) 39 (7%)	100 (25.5%) 274 (69.9%) 14 (3.6%)	102 (62.9%) 35 (21.6%) 25 (15.4%)	<0.0001 ^1,^*

MCT, monochemotherapy; LV, leucovorin; 5-FU, 5-fluorouracil; ^1^ Chi-square test *p*-value; * significant difference.

**Table 3 medicina-59-00350-t003:** Disease control rate and rate of response for patients with mCRC treated with bevacizumab and chemotherapy in the first and second lines.

	Bevacizumab/Oxaliplatin-Based Chemotherapy	Bevacizumab/Irinotecan-Based Chemotherapy	*p*-Value
First-Line Therapy			
Patients (N)	274	100	
Patients evaluable for response (N)	172	62	
PR (N)	50	17	
CR (N)	7	4	
SD (N)	117	41	
PD (N)	100	38	
RR (PR + CR) (%)	20.8%	21%	0.421
DCR (PR + SD + CR) (%)	63.5%	62%	0.979
Second-Line Therapy			
Patients (N)	35	102	
Patients evaluable for response (N)	15	56	
PR (N)	2	12	
SD (N)	12	43	
CR (N)	1	1	
PD (N)	20	47	
RR (PR + CR) (%)	8.6%	12.6%	0.023
DCR (CR + PR + SD) (%)	42.9%	54.4%	0.530

PR—partial response, N—number, SD—stable disease, PD—progressive disease, RR—response rate, and DCR—disease control rate.

**Table 4 medicina-59-00350-t004:** (**a**) A univariate and multivariate analysis of the prognostic factors of PFS with bevacizumab in the first line. (**b**) A univariate and multivariate analysis of the prognostic factors of OS with bevacizumab in the first line.

(a)					
Factors		Univariate Analysis		Multivariate Analysis	
		Hazard Ratio	*p*-Value	Hazard Ratio	*p*-Value
Age		1.01 (0.986–1.016)	0.883		
Cancer grading	G1 vs. G3 G2 vs. G3	0.91 (0.45–1.81) 0.92 (0.48–1.77)	0.782 0.802		
Resection status of the primary tumor	yes vs. no	0.84 (0.54–1.28)	0.407		
Location of metastases	liver vs. other	1.089 (0.73–1.62)	0.673		
Location of the tumor	right vs. left	1.13 (0.8–1.58)	0.499		
Chemotherapy regimen	Irinotecan vs. oxali-based Fluoropy vs. oxali-based	0.92 (0.63–1.35) 1.11 (0.57–2.16)	0.666 0.759		
RAS status		0.88 (0.75–1.04)	0.126		
(**b**)					
**Factors**		**Univariate Analysis**		**Multivariate Analysis**	
		**Hazard Ratio**	***p*-Value**	**Hazard Ratio**	***p*-Value**
Age		0.77 (0.51–1.18)	0.437	1.02 (1–1.03)	0.46
Cancer grading	G1 vs. G3 G2 vs. G3	0.91 (0.46–1.81) 1.09 (0.57–2.09)	0.782 0.799		
Resection status of the primary tumor	yes vs. no	1.02 (1.003–1.03)	0.016		
Location of metastases	liver vs. other	1.14 (0.77–1.69)	0.524		
Location of the tumor	right vs. left	1.54 (1.09–2.16)	0.014	1.42 (1.0–2.02)	0.047
Chemotherapy regimen	Irinotecan vs. oxali-based Fluoropy vs. oxali-based	0.87 (0.59–1.28) 1.34 (0.69–2.61)	0.474 0.391		
RAS status		0.88 (0.75–1.03)	0.114		

**Table 5 medicina-59-00350-t005:** Bevacizumab-related adverse events in the study population.

Adverse Event	Whole Group (N = 537) *		Bevacizumab in First Line (N = 382) *		Bevacizumab in Second Line (N = 155) *	
	Grade 1–2	Grade 3–4	Grade 1–2	Grade 3–4	Grade 1–2	Grade 3–4
Hypertension	75 (14%)	11 (2%)	48 (13%)	7 (1.8%)	27 (17%)	4 (2.6%)
Proteinuria	64 (12%)	2 (0.4%)	40 (10%)	1 (0.2%)	24 (15%)	1 (0.6%)
Bleeding	134 (25%)	5 (0.9%)	80 (21%)	3 (0.8%)	54 (34%)	2 (1.3)
Venous thromboembolic events	27 (5%)	11 (2%)	18 (4.7%)	6 (1.6%)	9 (5.8%)	5 (3.2%)
Arterial thromboembolic events	11 (2%)	5 (0.9%)	6 (1.6%)	3 (0.8%)	5 (3.2%)	2 (1.3%)
Gastrointestinal perforation	3 (0.5%)	6 (1%)	3 (0.8%)	4 (1%)	0	2 (1.3%)
Wound healing complications	11 (2%)	2 (0.3%)	11 (2.8%)	2 (0.5%)	0	0
Fistula	8 (1.5%)	7 (1.3%)	5 (1.3%)	6 (1.5%)	3 (2%)	1 (0.6%)

* safety data for seventeen patients are missing (ten patients received bevacizumab in the first line and seven in the second line.

## Data Availability

The data presented in this study are available on request from the corresponding author.

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
