# Peer review of "Bevacizumab Treatment for Metastatic Colorectal Cancer in Real-World Clinical Practice"

_medicina, 2023, doi:10.3390/medicina59020350_

Round 1

Reviewer 1 Report

General

An interesting, well present study about systemic therapy of metastasizing colorectal carcinoma. The authors present important, clinically relevant and in some parts also new data.

One major question arises when reading the manuscript: Represents this paper really a “real world study”? Data come from two central university clinics. We do not read in the paper how many patients come from “remote regions” or suffer from comorbidities – as the authors correctly state at line 61 of their manuscript as typical characteristics of “real world settings”. Age is mentioned, but it does not strongly differ from that in prospective clinical studies, and comorbidities are not mentioned.

The authors should discuss this point.

Minor points

Line 77: What means Bevacizumab administration was “delayed”?

Line 83: The authors collected data of death (“date,cause”). Cause of death is not shown at the results chapter.

Author Response

Thank you for your feedback and thoughtful comments.

Reviewer 2 Report

The study from Ioana Mihaela Dinu et al provided the outcomes and safety profile of bevacizumab for mCRC in a real-world study. This study enabled us to have a better understanding for the application of bevacizumab in real-world settings other than randomized controlled trials. Overall, this study was promising and with dramatical clinical significance. I have several issues needs to be addressed.

1. In the discussion part, the comparison between this real-world study and other real-world studies was missing. Was this study the only real-world one for bevacizumab in mCRC patients?

2. The comparison between this real-world study and other reported studies could be added. For example, in line 232 “The safety profile of bevacizumab was generally as expected. Grade 3-4 AE related to bevacizumab were reported for 8.9% of patients, similar to the results of other observational studies[13] ”. Here only one paper was cited, I did not think this one was the only one study with safety data for bevacizumab in mCRC patients. In addition, how about the Grade 1-2 AE as sompared with other studies.

3. In the Table 3, the RR% and DCR% was higher in Bevacizumab/irinotecan-based chemotherapy group from second line therapy. Was this data statistically significant. Authors should add the statistcs in Table 3.

4. In line 106-109, authors indicated that “Univariate analysis was used to identify the prognostic factors for survival in first-line and in second-line. The factors with a p-value less than 0.05 were incorporated into the subsequent multivariable analysis by Cox proportional hazards regression model” in the statistical part. However, none of univariate and  multivariable analysis results were presented in this paper. They should add these data.

Author Response

(The authors gave the same response as above.)
